# Associations between participation in organised physical activity in the school or community outside school hours and neighbourhood play with child physical activity and sedentary time: a cross-sectional analysis of primary school-aged children from the UK

Russell Jago,[1] Corrie Macdonald-Wallis,[1] Emma Solomon-Moore,[1] Janice, L. Thompson,[2] Debbie, A. Lawlor,[3,4] Simon, J. Sebire[1]

For numbered affiliations see end of article.

**Correspondence to**
Professor Russell Jago;
russ.jago@bris.ac.uk

## ABSTRACT

**Objectives** To assess the extent to which participation in organised physical activity in the school or community outside school hours and neighbourhood play was associated with children's physical activity and sedentary time.

**Design** Cross-sectional study.

**Setting** Children were recruited from 47 state-funded primary schools in South West England.

**Participants** 1223 children aged 8–9 years old.

**Outcome measures** Accelerometer-assessed moderate-to-vigorous-intensity physical activity (MVPA) and sedentary time.

**Methods** Children wore an accelerometer, and the mean minutes of MVPA and sedentary time per day were derived. Children reported their attendance at organised physical activity in the school or community outside school hours and neighbourhood play using a piloted questionnaire. Cross-sectional linear and logistic regression were used to examine if attendance frequency at each setting (and all settings combined) was associated with MVPA and sedentary time. Multiple imputation methods were used to account for missing data and increase sample size.

**Results** Children who attended clubs at school 3–4 days per week obtained an average of 7.58 (95% CI 2.7 to 12.4) more minutes of MVPA per day than children who never attended. Participation in the three other non-school-based activities was similarly associated with MVPA. Evidence for associations with sedentary time was generally weaker. Associations were similar in girls and boys. When the four different contexts were combined, each additional one to two activities participated in per week increased participants' odds (OR: 1.18, 95% CI 1.12 to 1.25) of meeting the government recommendations for 60 min of MVPA per day.

**Conclusion** Participating in organised physical activity at school and in the community is associated with greater physical activity and reduced sedentary time among both boys and girls. All four types of activity contribute to overall

### Strengths and limitations of this study

► Accelerometer data from a large sample of year 4 children.
► Detailed information on organised physical activity in the school or community outside school hours and neighbourhood play.
► Multiple imputation models to provide estimates for participants with missing data.
► Cross-sectional study design.
► Data are from a single UK region.

physical activity, which provides parents with a range of settings in which to help their child be active.

### INTRODUCTION

Physical activity is associated with improved mental well-being, reduced risk of obesity and lower blood pressure among children.[1] Sedentary time may also be a risk factor for non-communicable diseases, but it is not clear if this effect is independent of physical activity.[2–4] The UK Chief Medical Officers recommend that all children and young people should engage in at least an hour per day of moderate-to-vigorous-intensity physical activity (MVPA) and limit sedentary time[5]; however, considerable proportions of children do not meet these guidelines.[6] For example, data from the nationally representative millennium cohort showed that only 51% of children 7–8 years old met the recommendation.[7] The amount of time children spend engaged in MVPA gradually declines

with age, while sedentary time increases.[6 8–11] Strategies to increase children's physical activity are needed.

The majority of interventions to increase children's physical activity have been delivered during school time.[12 13] These interventions have included strategies such as changes to the physical education provision and new educational programmes based on information sharing and personal goal setting.[12–14] Overall, these programmes have tended to report no effect, weak effects or moderate effects in subgroups.[12–16] Potential reasons for these are the difficulty faced in adding interventions to already full school-curricula and the lack of skills and training that teachers have for delivering a range of activities to engage the majority of children.[17] As such, there is a need to understand the potential of organised physical activity outside school hours to increase MVPA.

After-school programmes have the potential to facilitate physical activity for children, as schools have space in which children can be active, staff who can be trained and many parents welcome programmes that provide child-care.[16 18–21] Although a number of studies have examined the potential of delivering such sessions, it is not clear whether attendance at the programmes currently provided by schools is associated with higher overall levels of MVPA.[18–21] There is also a lack of information about how attendance at community-based physical activity clubs contributes to overall MVPA. Furthermore, as not all children attend after-school programmes, it is not clear how other activities such as playing in the neighbourhood or at home in the garden contribute to overall MVPA. A key question, therefore, is whether the frequency of participation in organised physical activity in the school or community after school hours, neighbourhood play or home play is associated with the MVPA and sedentary time of children. Since some children will be active in all four settings, it would also be informative to examine collective participation across all settings.

The aim of this study was to assess among children (8–9 years of age) the extent to which participation in organised physical activity in the school or community outside school hours, and playing with friends or family near the home or in the garden, were associated with MVPA and sedentary time. A secondary aim was to examine if there was a cumulative association between participation in the four different types of activities with both MVPA and sedentary time.

## METHODS

The current analyses used data from the B-Proact1v study, which has been described in detail elsewhere.[11 22 23] Briefly, the study aimed to examine physical activity behaviours of children and their parents over the course of primary school. Between 2012 and 2013, data were collected from 1299 year 1 children (5–6 years of age) from 57 schools in Bristol (UK). Between March 2015 and July 2016, all 57 schools were approached to rejoin the study when the children were in year 4 (8–9 years of age), with 47 schools

agreeing to take part (1223 children). The current analyses used data from the year 4 assessments. The study received ethical approval from the School for Policy Studies Ethics Committee at the University of Bristol, and written parent consent was received from all participants.

### Data collection

Data were collected at schools, with children asked to complete a brief questionnaire. As indicators of organised physical activity outside school hours in school and in the community, respectively, we asked 'How often do you attend…a) a sport or exercise club at school (NOT including PE)? and b) a sport or exercise club at places other than your school (like a football club or ballet)?' To indicate neighbourhood play outside and within the home we asked 'How much do you play with your friends and family… a) outside near your home? and b) in your home or garden?' These questions each had four response options: 'Never', '1–2 days per week', '3–4 days per week' or '5 days per week'. We assigned these 0, 1, 2 and 3 points, respectively, and summed responses to derive an overall activity score ranging from 0 to 12, with a higher value indicating a higher frequency of activity participation.

Child height was measured to the nearest 0.1 cm using a SECA Leicester stadiometer (HAB International, Northampton, UK). Weight was recorded to the nearest 0.1 kg using a SECA 899 digital scale (HAB International). Child body mass index (BMI=kg/m$^2$) was then calculated and converted to an age-specific and gender-specific SD score.[24 25] Children wore a waist-worn ActiGraph wGT3X-BT accelerometer for 5 days including two weekend days. Parents provided demographic information via a questionnaire, including child gender and date of birth. Where children's date of birth was missing (20.5% of children), they were assigned the median age of 9.0 years. Indices of Multiple Deprivation (IMD) scores, based on the English Indices of Deprivation (http://data.gov.uk/dataset/index-of-multiple-deprivation), were assigned to each child based on their reported home postcode, where higher IMD scores indicate a greater level of deprivation.

### Accelerometer data reduction

Accelerometer data were processed using Kinesoft (V.3.3.75; Kinesoft, Saskatchewan, Canada). At least three valid days of data were required for accelerometer data to be considered complete for a given child and included in analysis, where a valid day was defined as at least 500 min of data, after excluding intervals of ≥60 min of zero counts, allowing up to 2 min of interruptions. We recognise that there is considerable variation in the number of minutes of accelerometer data that are required to be considered representative of a valid day.[26] These have ranged from 360 min per day, which has been used for children 6–8 years old,[27] to 800 min, which has been used for older children.[28 29] Within the field there is no consensus on the minimum number of minutes per

day that are needed for a day to be considered valid. We, therefore, adopted a 500 min per day threshold to ensure that our data are comparable to the methods employed by the International Children's Accelerometer Database,[6] which has pooled data from over 27 000 children across 20 large global cohorts. The child's average number of sedentary and MVPA minutes per day were derived using population-specific cut points for children.[30] We also derived a binary variable indicating whether the child's average daily MVPA was greater than the 60 min per day recommended by the UK government.[5]

## Analysis

The associations of child characteristics (gender, age, BMI z-score and IMD) with activity participation were examined in the observed data using t-tests, Pearson's correlation coefficients, $X^2$ tests and one-way analysis of variance as appropriate.

Multiple imputation of missing data was used to create 20 imputed data sets for the 1223 year 4 children. We used 20 cycles of regression switching and combined regression coefficients across data sets using Rubin's rules.[31] We imputed separately for boys and girls to allow for associations to differ by gender. All exposures (organised physical activity attendance and neighbourhood play), outcomes (sedentary time and MVPA), potential confounders (gender, age, BMI z-score and IMD) and child's school were included in multiple imputation models, and achievement of the MVPA guideline and overall activity score was imputed passively. Any children with less than three valid days of accelerometer data had their accelerometer measures imputed.

We examined the pairwise associations of the activity participation variables by dichotomising, cross-tabulating and fitting unadjusted logistic regression models of one frequency variable on another.

We used linear regression models to examine the associations of activity participation and the overall activity score with the child's average sedentary and MVPA minutes per day, and logistic regression models to examine associations with achievement of the MVPA guideline. In model 1 we adjusted for gender and age, and in model 2 we adjusted additionally for BMI z-score and IMD. To account for the clustering of children in schools and the associated potential to underestimate the SEs which are used to compute the 95% CIs and p values, robust SEs, which took account of the school level clustering, were used for all models. Combined Wald tests were used to test for evidence of interaction between the child's gender and the exposure of interest.

We predicted the children's mean number of sedentary and MVPA minutes per day by frequency of participation in each activity based on linear combinations of the regression coefficients from fully adjusted models (model 2).

Regression analyses were repeated restricting to the children who had complete data for all exposures, outcomes and covariables, and compared with the

multiple imputation analysis. We also produced scatter plots of sedentary time and MVPA by the overall activity score in the observed data.

A sensitivity analysis was performed including accelerometer data for any children who had at least one valid day of measures to assess whether only including accelerometer data for children who recorded at least three valid days influenced our results. All analyses were performed on Stata V.14.0.

## RESULTS

The distributions of characteristics of the children in the observed data, multiple imputation data sets and subset with complete information are shown in table 1. All characteristics showed similar distributions in each of the data sets and had only a small proportion of missing data (maximum 16.1% for accelerometer measures).

BMI, body mass index; IMD, Indices of Multiple Deprivation; MVPA, moderate-to-vigorous-intensity physical activity.

### Associations of child characteristics with activity participation

Online supplementary table S1 shows child activity attendance by gender. Girls tended to report lower frequencies of participating in organised physical activity in the school or community outside school hours, and had a lower mean overall activity score. There was no gender difference in friends/family play either in or outside of the home. The associations of other child characteristics with activity participation are shown in online supplementary table S2. There was some evidence that children who more frequently attended a sport/exercise club outside of school had a higher mean age. Child BMI was not strongly associated with any particular activity, but there was weak evidence that the overall activity score decreased with increasing BMI z-score. Children who reported attendance of 'Never' or '5 days/week' generally had the highest IMD scores, suggesting a U-shaped association, and there was a negative correlation between the overall activity score and IMD.

### Inter-relationships of activity participation frequencies

Participating in one type of activity more frequently was generally associated with a higher frequency of participation in each of the others, except that attendance at a sport/exercise club outside of school was not associated with playing outside near the home (online supplementary table S3).

### Associations of activity participation with sedentary time and MVPA

There was a negative correlation of the overall activity score with sedentary time and a positive correlation between the overall activity score and MVPA (online supplementary figures S1 and S2).

**Table 1** Characteristics of children who took part in the year 4 phase of the B-Proact1v study (observed and multiple imputation) (n=1223)

| Child characteristics | | Observed data | | Imputed data (n=1223) | Complete data (n=987) |
| --- | --- | --- | --- | --- | --- |
| | | N available | Mean (SD) or % | Mean (SD) or % | Mean (SD) or % |
| Sedentary time at year 4 (min/day) | | 1026 | 445.4 (115.4) | 444.7 (120.1) | 446.0 (116.9) |
| MVPA at year 4 (min/day) | | 1026 | 61.6 (21.9) | 61.9 (22.5) | 61.8 (21.8) |
| Met MVPA guidelines at year 4 | No | 1163 | 53.2 | 52.6 | 53.4 |
| | Yes | | 46.8 | 47.4 | 46.6 |
| Gender | Boy | 1223 | 45.5 | 45.5 | 44.5 |
| | Girl | | 54.5 | 54.5 | 55.5 |
| Age at year 4 (years) | | 1223 | 9.03 (0.41) | 9.03 (0.41) | 9.03 (0.43) |
| BMI z-score at year 4 | | 1208 | 0.35 (1.08) | 0.36 (1.08) | 0.31 (1.07) |
| IMD score at year 4 | | 1204 | 15.9 (14.1) | 15.9 (14.2) | 15.3 (13.6) |
| Frequency child attends sport/ exercise club at school | Never | 1215 | 27.8 | 27.9 | 28.2 |
| | 1–2 days per week | | 45.5 | 45.5 | 45.8 |
| | 3–4 days per week | | 16.1 | 16.1 | 16.6 |
| | 5 days per week | | 10.5 | 10.5 | 9.4 |
| Frequency child attends sport/ exercise club outside school | Never | 1214 | 20.6 | 20.6 | 19.4 |
| | 1–2 days per week | | 50.2 | 50.2 | 51.0 |
| | 3–4 days per week | | 20.8 | 20.9 | 21.5 |
| | 5 days per week | | 8.3 | 8.3 | 8.2 |
| Frequency child plays with friends/ family outside near home | Never | 1205 | 6.3 | 6.4 | 6.5 |
| | 1–2 days per week | | 33.7 | 33.7 | 34.3 |
| | 3–4 days per week | | 29.0 | 28.9 | 29.7 |
| | 5 days per week | | 31.0 | 30.9 | 29.5 |
| Frequency child plays with friends/ family in home or garden | Never | 1199 | 9.6 | 9.7 | 8.8 |
| | 1–2 days per week | | 34.5 | 34.6 | 35.0 |
| | 3–4 days per week | | 26.8 | 26.7 | 27.8 |
| | 5 days per week | | 29.1 | 29.0 | 28.5 |
| Activity frequency score | | 1193 | 5.88 (2.29) | 5.86 (2.29) | 5.84 (2.26) |

Table 2 shows the mean difference in sedentary time by activity participation and overall activity score in the multiple imputation data sets, and figure 1 shows the predicted sedentary time by activity participation. Sedentary time decreased on average with increasing frequency of attending sport/exercise clubs either at school or outside of school and with increasing frequency of playing with friends/family outside near the home in regression models adjusted for gender and age (model 1). The association between sport/exercise club attendance outside of school and sedentary time weakened slightly on additional adjustment for BMI z-score and IMD (model 2), but other associations remained. An increase in children's overall activity score was also strongly associated with a reduction in sedentary time in both models. However, there was no evidence of an association between playing with friends/family at home and sedentary time. Associations did not differ between boys and girls. Findings were similar when restricted to children who had complete data (online supplementary table S4).

The mean difference in MVPA by each of the activity variables in the multiple imputation data is shown in table 3, with predicted MVPA by activity participation presented in figure 2. Higher frequencies of attending sport/exercise clubs either at school or outside of school and of play either outside or in the home/garden were all associated with greater MVPA on average in models 1 and 2. Associations were similar in boys and girls. A higher overall activity score was also associated with greater MVPA, with some evidence that this association was stronger in boys than in girls. Associations were similar when restricted to children with complete data (online supplementary table S5).

The associations of activity variables with achievement of the hour per day government guideline in the multiple

**Table 2** Mean difference in the children's average sedentary minutes per day associated with different activities using multiple imputation (n=1223)*

| Exposure | Sedentary time (min/day): mean difference (95% CI) | | | | | | p for gender interaction |
| | All (n=1223) | | Boys (n=556) | | Girls (n=667) | | |
| | Model 1 | Model 2 | Model 1 | Model 2 | Model 1 | Model 2 | |
|---|---|---|---|---|---|---|---|
| **Frequency child attends sport/exercise club at school** | | | | | | | |
| Never (ref) | 0 | 0 | 0 | 0 | 0 | 0 | 0.69 |
| 1–2 days/week | −4.2 (−22.5 to 14.0) | −3.0 (−21.4 to 15.4) | 7.9 (−16.6 to 32.3) | 8.9 (−15.9 to 33.7) | −13.0 (−38.3 to 12.4) | −11.5 (−37.5 to 14.4) | |
| 3–4 days/week | −31.1 (−50.7 to 11.5) | −28.8 (−46.4 to 11.3) | −18.7 (−41.8 to 4.4) | −16.9 (−41.2 to 7.3) | −40.1 (−71.7 to 8.5) | −37.3 (−65.4 to −9.2) | |
| 5 days/week | −18.7 (−42.9 to 5.6) | −18.8 (−42.1 to 4.5) | −9.6 (−37.7 to 18.4) | −7.3 (−34.9 to 20.4) | −26.0 (−66.8 to 14.8) | −33.3 (−82.0 to 15.5) | |
| p for trend | 0.02 | 0.01 | 0.17 | 0.23 | 0.05 | 0.04 | |
| **Frequency child attends sport/exercise club outside of school** | | | | | | | |
| Never (ref) | 0 | 0 | 0 | 0 | 0 | 0 | 0.81 |
| 1–2 days/week | 14.8 (−9.1 to 38.7) | 18.0 (−8.9 to 45.0) | 21.5 (−7.4 to 50.5) | 25.9 (−4.5 to 56.4) | 11.0 (−19.9 to 42.0) | 13.6 (−20.5 to 47.7) | |
| 3–4 days/week | −7.2 (−27.0 to 12.6) | −1.5 (−23.4 to 20.4) | −1.0 (−32.8 to 30.8) | 4.8 (−27.1 to 36.6) | −10.5 (−30.5 to 9.5) | −4.1 (−27.3 to 19.0) | |
| 5 days/week | −19.9 (−41.1 to 1.2) | −15.5 (−37.7 to 6.8) | −12.6 (−42.4 to 17.1) | −7.7 (−36.5 to 21.1) | −25.2 (−55.0 to 4.6) | −20.6 (−52.1 to 11.0) | |
| p for trend | 0.02 | 0.07 | 0.12 | 0.16 | 0.07 | 0.23 | |
| **Frequency child plays with friends/family outside near home** | | | | | | | |
| Never (ref) | 0 | 0 | 0 | 0 | 0 | 0 | 0.56 |
| 1–2 days/week | 0.77 (−28.5 to 30.0) | 2.3 (−28.0 to 32.7) | 19.2 (−16.5 to 55.0) | 21.2 (−14.4 to 56.8) | −16.6 (−63.1 to 30.0) | −15.6 (−65.0 to 33.9) | |
| 3–4 days/week | −5.2 (−29.9 to 19.6) | −2.8 (−29.4 to 23.7) | 7.6 (−28.4 to 43.5) | 10.6 (−25.5 to 46.6) | −17.6 (−65.8 to 30.5) | −16.1 (−67.4 to 35.3) | |
| 5 days/week | −23.4 (−49.0 to 2.1) | −24.1 (−50.8 to 2.7) | −11.0 (−45.2 to 23.2) | −10.2 (−44.4 to 24.0) | −35.4 (−79.9 to 9.0) | −37.8 (−86.6 to 10.9) | |
| p for trend | 0.004 | 0.001 | 0.07 | 0.05 | 0.02 | 0.02 | |
| **Frequency child plays with friends/family in home/garden** | | | | | | | |
| Never (ref) | 0 | 0 | 0 | 0 | 0 | 0 | 0.90 |
| 1–2 days/week | 16.2 (−13.8 to 46.3) | 17.9 (−12.5 to 48.2) | 19.2 (−22.9 to 61.3) | 20.6 (−21.5 to 62.7) | 13.1 (−25.8 to 52.0) | 14.9 (−24.3 to 54.1) | |
| 3–4 days/week | −3.9 (−28.8 to 20.9) | −0.7 (−26.7 to 25.4) | 3.9 (−30.5 to 38.4) | 6.5 (−28.7 to 41.7) | −10.7 (−46.6 to 25.3) | −6.4 (−42.9 to 30.1) | |
| 5 days/week | 5.3 (−22.9 to 33.5) | 6.1 (−22.5 to 34.7) | 9.9 (−32.5 to 52.3) | 11.6 (−31.2 to 54.4) | 1.2 (−39.2 to 41.5) | 0.6 (−40.2 to 41.5) | |
| p for trend | 0.49 | 0.49 | 0.85 | 0.91 | 0.49 | 0.45 | |
| Activity score (per unit) | −4.8 (−7.3 to 2.3) | −4.6 (−7.0 to 2.1) | −3.8 (−7.7 to 0.1) | −3.4 (−7.2 to 0.4) | −5.9 (−10.0 to 1.7) | −5.9 (−10.2 to −1.6) | 0.41 |

*Model 1 is adjusted for age and gender; model 2 is additionally adjusted for body mass index and Indices of Multiple Deprivation score.

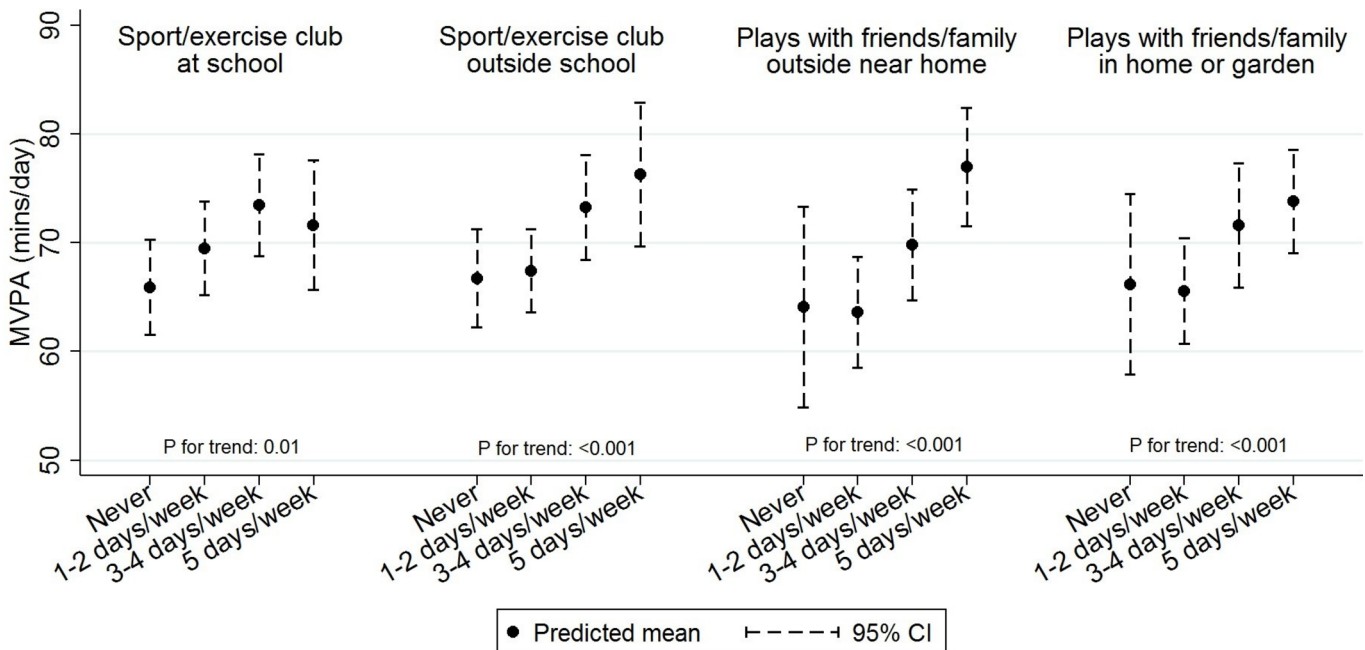

**Figure 1** Predicted time spent in sedentary behaviour by type of activity using multiple imputation (n=1223). Predictions were obtained from fully adjusted regression models (model 2) including all year 4 children (boys and girls) and are for a 9-year-old boy with a BMI z-score of 0 and IMD score of 16. Predicted sedentary time for girls was approximately 15–17 min/day higher (same additive effect across all categories of the exposure variable) depending on the regression model. BMI, body mass index; IMD, Indices of Multiple Deprivation; MVPA, moderate-to-vigorous-intensity physical activity.

imputation data are shown in table 4. A higher frequency of participation in any of the activities, or an increase in the overall activity score, was associated with increased odds of meeting the government guideline in both models. Associations were similar in girls and boys. A unit increase in the activity score was associated with around an 18% increase in the odds of achieving 60 min of MVPA per day. Findings were unchanged when restricting to those with complete data (online supplementary table S6).

### Sensitivity analysis

When we changed the inclusion criteria for accelerometer measures from three to one valid day, findings were largely unchanged, except that boys showed a stronger association between sport/exercise club attendance at their school and MVPA than girls (p for interaction=0.03 in multiple imputation data).

### DISCUSSION

The findings demonstrate that increased participation in organised physical activity at school and in the community is associated with greater overall physical activity and reduced sedentary time among both boys and girls. Specifically, a child who attends a school-based club 3–4 days per week obtained 7.8 more minutes of MVPA per day than a child who did not attend at all, with attendance of 5 days a week at a sport/exercise club outside of school associated with 9.9 more minutes of MVPA than children who never attended clubs. There were comparable patterns for engagement in non-organised activity at home or in

the neighbourhood—both were associated with increased MVPA—but only activity outside of the home was associated with reduced sedentary time. When the four different contexts of physical activity were combined, the analyses showed that each additional one to two activities participated in per week increased the odds of meeting the Chief Medical Officers' recommendation of 60 min of MVPA per day by 18%. Thus, encouraging children to attend after-school and community-based physical activity clubs, as well as to play at home and in the neighbourhood, is critical for helping children to increase MVPA. Moreover, in light of the relative consistency in findings for each of the four forms of physical activity, the message to parents should be that physical activity can be accumulated in all four settings, which allows them to find a balance that works for their family. For some families with working parents, after-school programmes may be the key activity to focus on, whereas for other families encouraging children to play in the neighbourhood is likely to be useful for maximising physical activity. Furthermore, as there was little evidence that play at home was associated with a reduction in sedentary time, it is also important to examine ways of encouraging non-sedentary activities within the home.

The findings in this paper support previous research that has shown that introducing extracurricular clubs into the school day can promote increased physical activity among primary school-aged children.[32] The study is in broad agreement with the body of work that has shown that risky outdoor play and higher independent mobility are associated with higher levels of physical

**Table 3** Mean difference in the children's average MVPA minutes per day associated with different activities using multiple imputation (n=1223)*

| Exposure | | MVPA (min/day): mean difference (95% CI) | | | | | | p for gender interaction |
|---|---|---|---|---|---|---|---|---|
| | | All (n=1223) | | Boys (n=556) | | Girls (n=667) | | |
| | | Model 1 | Model 2 | Model 1 | Model 2 | Model 1 | Model 2 | |
| Frequency child attends sport/exercise club at school | Never (ref) | 0 | 0 | 0 | 0 | 0 | 0 | 0.64 |
| | 1–2 days/week | 3.7 (0.3 to 7.2) | 3.6 (0.1 to 7.0) | 4.4 (−2.1 to 10.9) | 4.1 (−2.3 to 10.5) | 3.2 (−0.3 to 6.7) | 3.1 (−0.3 to 6.6) | |
| | 3–4 days/week | 7.8 (2.9 to 12.6) | 7.6 (2.7 to 12.4) | 10.1 (2.6 to 17.7) | 9.6 (2.0 to 17.2) | 5.9 (0.9 to 10.8) | 5.8 (0.9 to 10.7) | |
| | 5 days/week | 5.9 (−0.1 to 11.8) | 5.7 (−0.3 to 11.7) | 7.5 (−0.4 to 15.4) | 7.0 (−1.1 to 15.0) | 3.8 (−3.7 to 11.3) | 3.6 (−4.1 to 11.4) | |
| p for trend | | 0.007 | 0.01 | 0.02 | 0.03 | 0.05 | 0.06 | |
| Frequency child attends sport/exercise club outside of school | Never (ref) | 0 | 0 | 0 | 0 | 0 | 0 | 0.80 |
| | 1–2 days/week | 0.9 (−2.5 to 4.3) | 0.7 (−2.9 to 4.4) | 1.8 (−4.7 to 8.3) | 1.3 (−5.5 to 8.2) | 0.5 (−3.3 to 4.4) | 0.5 (−3.4 to 4.5) | |
| | 3–4 days/week | 6.9 (2.4 to 11.5) | 6.5 (1.9 to 11.2) | 8.9 (1.0 to 16.8) | 8.1 (0.1 to 16.0) | 5.5 (0.9 to 10.1) | 5.6 (1.1 to 10.2) | |
| | 5 days/week | 9.9 (3.8 to 16.0) | 9.6 (3.3 to 15.9) | 11.8 (3.4 to 20.1) | 11.2 (2.4 to 19.9) | 8.2 (0.7 to 15.7) | 8.2 (0.8 to 15.7) | |
| p for trend | | <0.001 | <0.001 | 0.001 | 0.002 | 0.007 | 0.006 | |
| Frequency child plays with friends/family outside near home | Never (ref) | 0 | 0 | 0 | 0 | 0 | 0 | 0.19 |
| | 1–2 days/week | 0.7 (−4.6 to 6.0) | 0.4 (−4.9 to 5.6) | 0.0 (−9.3 to 9.2) | −0.5 (−9.9 to 8.9) | 1.4 (−6.0 to 8.8) | 1.3 (−6.2 to 8.7) | |
| | 3–4 days/week | 5.1 (−0.9 to 11.1) | 4.8 (−1.1 to 10.7) | 6.1 (−4.1 to 16.3) | 5.7 (−4.5 to 15.9) | 4.4 (−3.5 to 12.3) | 4.3 (−3.7 to 12.2) | |
| | 5 days/week | 9.6 (4.0 to 15.3) | 9.5 (3.9 to 15.2) | 13.0 (3.3 to 22.8) | 12.9 (3.0 to 22.7) | 6.7 (−0.7 to 14.0) | 6.6 (−0.9 to 14.0) | |
| p for trend | | <0.001 | <0.001 | <0.001 | <0.001 | 0.005 | 0.006 | |
| Frequency child plays with friends/family in home/garden | Never (ref) | 0 | 0 | 0 | 0 | 0 | 0 | 0.38 |
| | 1–2 days/week | 2.0 (−2.4 to 6.4) | 1.6 (−2.7 to 6.0) | −0.1 (−8.4 to 8.3) | −0.6 (−8.9 to 7.7) | 3.8 (−2.2 to 9.8) | 3.6 (−2.4 to 9.7) | |
| | 3–4 days/week | 5.6 (0.9 to 10.3) | 5.3 (0.4 to 10.2) | 5.6 (−3.1 to 14.3) | 5.4 (−3.5 to 14.4) | 5.7 (−0.6 to 11.9) | 5.5 (−0.9 to 12.0) | |
| | 5 days/week | 7.4 (3.0 to 11.8) | 7.1 (2.7 to 11.6) | 8.3 (0.1 to 16.6) | 7.7 (−0.7 to 16.1) | 6.6 (0.7 to 12.5) | 6.5 (0.6 to 12.5) | |
| p for trend | | <0.001 | <0.001 | 0.004 | 0.006 | 0.02 | 0.02 | |
| Activity score (per unit) | | 2.1 (1.4 to 2.7) | 2.0 (1.4 to 2.7) | 2.6 (1.7 to 3.4) | 2.5 (1.6 to 3.4) | 1.6 (0.8 to 2.3) | 1.6 (0.8 to 2.3) | 0.06 |

*Model 1 is adjusted for age and gender; model 2 is additionally adjusted for body mass index and Indices of Multiple Deprivation score.
MVPA, moderate-to-vigorous-intensity physical activity.

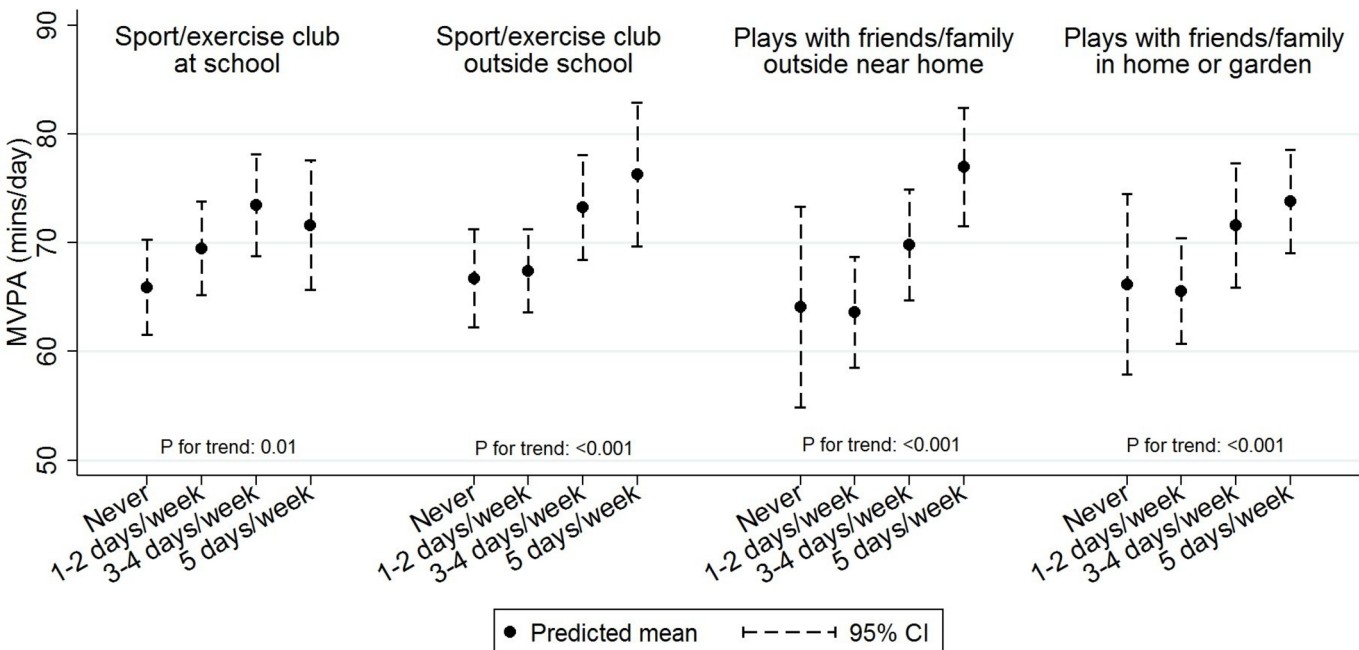

**Figure 2** Predicted time spent in MVPA by type of activity using multiple imputation (n=1223).* Predictions were obtained from fully adjusted regression models (model 2) including all year 4 children (boys and girls) and are for a 9-year-old boy with a BMI z-score of 0 and IMD score of 16. Predicted time spent in MVPA for girls was approximately 13 min/day lower (same additive effect across all categories of the exposure variable). BMI, body mass index; IMD, Indices of Multiple Deprivation; MVPA, moderate-to-vigorous-intensity physical activity.

activity among children.[33–36] These findings complement these bodies of work by showing an additional effect of accumulating physical activity across different settings to maximise the overall amount of physical activity in which children engage.

The UK Childhood Obesity strategy recommends that all primary schools should provide at least 30 min per day of physical activity opportunities across the curriculum, break times and extracurricular activities.[37] The data presented here show that in the imputed data set, 72.2% of year 4 children were attending school-based programmes at least once per week and 10.5% were attending 5 days per week. Previous research has shown that in the UK, after-school clubs for primary school children tend to be dominated by team sports, such as football and rugby, with limited provision for non-competitive physical activities.[38] Thus, increasing the number and variety of sessions that children attend and improving the quality of those sessions are likely to provide a cost-effective means of increasing children's physical activity. This hypothesis is consistent with the recent theory of expanded, extended and enhanced opportunities, which suggests that the most effective means of increasing children's physical activity will be provided by extending and expanding current provision.[19] Thus, schools and community groups should be encouraged to extend current after-school provision to more children, diversify the activities to interest more pupils (preferably involving pupils in deciding what activities to offer) and enhance the quality of provision to maximise the amount of activity obtained. These relatively simple changes could be made at each school and

would provide scalable ways for increasing overall levels of physical activity and contributing to the UK government's goal of reducing the prevalence of childhood obesity.

### Strengths and limitations
The major strength of this study is the large sample size and provision of detailed information about participation in four different physical activity settings alongside accelerometer-assessed physical activity. In addition, the use of multiple imputation models to provide estimates for participants with missing data using a robust methodology has enabled us to maximise the sample for analysis. The study is limited by the cross-sectional design, which limits the ability to infer causation between frequency of participation in different settings and levels of physical activity. All questions were self-reported, and it is possible that some were recalled more accurately than others. Moreover, as the questions used were developed for this project, we do not have information on the reliability and validity of the scale. The report of play within the home is likely to include both sedentary and physically active forms of play, as the question included play indoors, which could be expected to be more sedentary, as well as outdoors in the garden, which is likely to be more active. Equally, as the question focused on play with friends or family, we do not have any information about individual play, and we were unable to disentangle these inter-related issues. We also cannot rule out the possibility of residual confounding, but have adjusted for several key potential confounding variables in order to minimise this. The study is also drawn from the greater Bristol area

**Table 4** ORs for achieving 60min of MVPA per day associated with different activities using multiple imputation (n=1223)*

| Exposure | | Meeting government guideline: OR (95% CI) | | | | | | p for gender interaction |
|---|---|---|---|---|---|---|---|---|
| | | All (n=1223) | | Boys (n=556) | | Girls (n=667) | | |
| | | Model 1 | Model 2 | Model 1 | Model 2 | Model 1 | Model 2 | |
| Frequency child attends sport/exercise club at school | Never (ref) | 1 | 1 | 1 | 1 | 1 | 1 | 0.34 |
| | 1–2 days/week | 1.33 (0.98 to 1.81) | 1.31 (0.97 to 1.78) | 1.10 (0.65 to 1.86) | 1.07 (0.63 to 1.81) | 1.54 (1.06 to 2.23) | 1.54 (1.07 to 2.21) | |
| | 3–4 days/week | 1.87 (1.16 to 3.01) | 1.83 (1.14 to 2.93) | 2.22 (1.09 to 4.53) | 2.14 (1.04 to 4.40) | 1.63 (0.94 to 2.85) | 1.63 (0.94 to 2.82) | |
| | 5 days/week | 1.69 (1.03 to 2.78) | 1.67 (1.02 to 2.76) | 1.61 (0.83 to 3.09) | 1.55 (0.79 to 3.02) | 1.76 (0.84 to 3.69) | 1.77 (0.84 to 3.73) | |
| p for trend | | 0.01 | 0.01 | 0.03 | 0.05 | 0.05 | 0.05 | |
| Frequency child attends sport/exercise club outside of school | Never (ref) | 1 | 1 | 1 | 1 | 1 | 1 | 0.93 |
| | 1–2 days/week | 1.14 (0.80 to 1.64) | 1.13 (0.77 to 1.64) | 1.21 (0.74 to 1.98) | 1.18 (0.70 to 1.99) | 1.11 (0.68 to 1.80) | 1.12 (0.69 to 1.82) | |
| | 3–4 days/week | 1.82 (1.22 to 2.72) | 1.75 (1.16 to 2.65) | 2.07 (1.11 to 3.87) | 1.95 (1.03 to 3.67) | 1.66 (0.99 to 2.78) | 1.69 (1.01 to 2.83) | |
| | 5 days/week | 2.70 (1.57 to 4.62) | 2.63 (1.51 to 4.58) | 3.24 (1.49 to 7.07) | 3.15 (1.41 to 7.06) | 2.30 (1.12 to 4.72) | 2.33 (1.14 to 4.77) | |
| p for trend | | <0.001 | <0.001 | <0.001 | 0.001 | 0.009 | 0.008 | |
| Frequency child plays with friends/family outside near home | Never (ref) | 1 | 1 | 1 | 1 | 1 | 1 | 0.74 |
| | 1–2 days/week | 1.15 (0.70 to 1.88) | 1.12 (0.69 to 1.82) | 0.98 (0.46 to 2.09) | 0.94 (0.43 to 2.06) | 1.47 (0.62 to 3.45) | 1.47 (0.63 to 3.43) | |
| | 3–4 days/week | 1.88 (1.12 to 3.16) | 1.84 (1.11 to 3.06) | 1.62 (0.73 to 3.58) | 1.59 (0.70 to 3.58) | 2.32 (0.91 to 5.91) | 2.32 (0.92 to 5.88) | |
| | 5 days/week | 2.11 (1.26 to 3.52) | 2.10 (1.26 to 3.49) | 2.08 (0.92 to 4.69) | 2.07 (0.91 to 4.74) | 2.35 (0.93 to 5.99) | 2.36 (0.93 to 6.02) | |
| p for trend | | <0.001 | <0.001 | 0.003 | 0.002 | 0.01 | 0.02 | |
| Frequency child plays with friends/family in home/garden | Never (ref) | 1 | 1 | 1 | 1 | 1 | 1 | 0.21 |
| | 1–2 days/week | 1.27 (0.84 to 1.91) | 1.23 (0.81 to 1.86) | 0.94 (0.52 to 1.72) | 0.90 (0.49 to 1.64) | 1.75 (0.84 to 3.64) | 1.75 (0.84 to 3.65) | |
| | 3–4 days/week | 1.51 (1.00 to 2.28) | 1.46 (0.96 to 2.23) | 1.29 (0.70 to 2.40) | 1.28 (0.68 to 2.44) | 1.80 (0.86 to 3.80) | 1.80 (0.86 to 3.79) | |
| | 5 days/week | 1.71 (1.14 to 2.56) | 1.67 (1.11 to 2.53) | 1.71 (0.90 to 3.25) | 1.62 (0.83 to 3.14) | 1.80 (0.88 to 3.69) | 1.81 (0.88 to 3.70) | |
| p for trend | | 0.005 | 0.006 | 0.01 | 0.02 | 0.21 | 0.21 | |
| Activity score (per unit) | | 1.18 (1.12 to 1.25) | 1.18 (1.11 to 1.25) | 1.21 (1.12 to 1.30) | 1.20 (1.11 to 1.30) | 1.16 (1.07 to 1.25) | 1.16 (1.07 to 1.25) | 0.52 |

*Model 1 is adjusted for age and gender; model 2 is additionally adjusted for body mass index and Indices of Multiple Deprivation score. MVPA, moderate-to-vigorous-intensity physical activity.

in the UK, and as such our ability to extend findings to other settings and countries is limited.

## CONCLUSIONS

Participation in organised physical activity at school and in the community is associated with greater physical activity and reduced sedentary time among both boys and girls. In light of the challenges of promoting physical activity during school time, parents should encourage children to attend after school clubs, attend community activity groups and play in the neighbourhood to help their children to meet physical activity guidelines. The data show that all four types of activity contribute similarly to overall physical activity, and there is therefore an opportunity for families to find the best mix of options for them.

**Author affiliations**
[1]Centre for Exercise, Nutrition and Health Sciences, School for Policy Studies, University of Bristol, Bristol, UK
[2]School of Sport, Exercise and Rehabilitation Sciences, University of Birmingham, Birmingham, UK
[3]MRC Integrative Epidemiology Unit at theUniversity of Bristol, University of Bristol, Bristol, UK
[4]Bristol Medical School, University of Bristol, Bristol, UK

**Twitter** @Russ_Jago

**Acknowledgements** We would like to thank all of the families and schools that have taken part in the B-Proact1v project. We would also like to thank all current and previous members of the research team who are not authors on this paper.

**Contributors** Conception/design: RJ, ES-M, JLT, DL and SS. Data analysis/ acquisition/interpretation: RJ, CM-W, ES-M and DL. Drafting/revising critically for important content: All authors. Final approval: All authors. Accountability for study and manuscript: RJ.

**Funding** This work was supported by grants from the British Heart Foundation (ref PG/11/51/28986 and SP 14/4/31123). DL works in a unit that receives funding from the University of Bristol and UK Medical Research Council (MC_UU_1201/5); she is also a UK National Institute of Health and Research Senior Investigator (NF-SI-0166-10196). The funders had no involvement in data analysis, data interpretation or writing of the paper.

**Competing interests** None declared.

**Patient consent** Parental/guardian consent obtained.

**Ethics approval** School for Policy Studies Ethics Committee at the University of Bristol.

**Provenance and peer review** Not commissioned; externally peer reviewed.

**Data sharing statement** The data sets generated during the current study are not publicly available as the project is ongoing and data are not ready for archiving. We will consider reasonable requests for access to the data once the project is complete in 2019.

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
