## [Reviewer comments · BMJ Open]

ARTICLE DETAILS

TITLE (PROVISIONAL)	Associations between participation in organised physical activity in the school or community outside school hours, and neighbourhood play with child physical activity and sedentary time: a cross-sectional analysis of primary school-aged children from the UK
AUTHORS	Jago, Russell; Macdonald-Wallis, Corrie; Solomon-Moore, Emma; Thompson, Janice; Lawlor, Debbie; Sebire, Simon

VERSION 1 - REVIEW

REVIEWER	Richard Larouche Healthy Active Living and Obesity Research Group, Children's Hospital of Eastern Ontario Research Institute. Canada
REVIEW RETURNED	30-May-2017

GENERAL COMMENTS	See below (note that in the questions above, I answered "no" for limitations - this relates to my comment about the reliability and validity of the questions used to measure the main exposures) Review of "Associations between participation in organized physical activity in the school or community outside school hours, and neighborhood play with children physical activity and sedentary time: a cross-sectional analysis" General comments: The authors examined different approaches that can be used to increase physical activity outside of school hours (organized sport/exercise at school or outside of school, and play with friends/family outside or in home/garden). The sample size is large and the authors have used multiple imputations to handle missing data. The finding that these 4 types of activities contribute to physical activity in a similar way is of particular interest as it can provide families with a range of options. That being said, I believe that some revisions are needed to strengthen the manuscript. Major comments: 1. Given the study design in which children were recruited in schools, I would have expected the authors to use multilevel
---

	models. (I realize that it is possible that the school-level intra-class correlation coefficient for MVPA and sedentary time was so low that the authors may have decided not to use multilevel models, but if this is the case, this should be specified in the manuscript). 2. In the discussion, the authors should compare their results with those of previous studies to a greater extent. For instance, there are many previous studies (even systematic reviews) that have looked at the types of activities considered in the manuscript, such as outdoor play and after-school programs. Minor comments: 1. Was the reliability and validity of the questions used to examine the type of activities assessed in this study or in a previous study? If yes, this should be mentioned in the methods. If not, then it should be mentioned as a limitation of the study. 2. Please justify the use of 500 minutes (8 hours and 20 minutes) as a threshold for inclusion of accelerometer data. I am asking this because it is well known that accelerometry data reduction methods vary substantially across studies which makes it difficult to compare the findings of different studies (see Cain et al. J Phys Act Health 2013;10(3):437-450 for detailed discussion of this issue). 3. The authors correctly acknowledge that they were not able to disentangle whether play in the home or garden occurred inside or outside due to the wording of the question. Another issue is that this question (and the one about play outside near the home) appeared to consider only play with friends and/or family. While there is evidence that friends/family can help encourage children to be active, it would be worthwhile to mention that the questions used do not allow the researchers to examine the contribution of play without friends and or family. Playing outside alone would likely involve more physical activity than playing inside alone.
--	---

REVIEWER	Erica Hinckson Auckland University of Technology, New Zealand
REVIEW RETURNED	06-Jun-2017

GENERAL COMMENTS	The authors examined the extent to which participation in organised physical activity in the school or community outside school hours, and neighbourhood play, were associated with children's physical activity and sedentary time This is a strong study with a large sample size with four reported physical activity settings and use of accelerometers to objectively measure PA and ST. All my concerns were directly addressed in the manuscript. It is an exceptional manuscript and ready for publication.
---

VERSION 1 – AUTHOR RESPONSE

Reviewer 1:

General comments: The authors examined different approaches that can be used to increase physical activity outside of school hours (organized sport/exercise at school or outside of school, and play with friends/family outside or in home/garden). The sample size is large and the authors have used multiple imputations to handle missing data. The finding that these 4 types of activities contribute to physical activity in a similar way is of particular interest as it can provide families with a range of options. That being said, I believe that some revisions are needed to strengthen the manuscript.

Response: Please find responses to each issue raised below.

Major comments:

1. Given the study design in which children were recruited in schools, I would have expected the authors to use multilevel models. (I realize that it is possible that the school-level intra-class correlation coefficient for MVPA and sedentary time was so low that the authors may have decided not to use multilevel models, but if this is the case, this should be specified in the manuscript).

Response: We recognise that our study consists of pupils within schools and that there are likely to be correlations between children from the same school in their activity levels that need to be taken into account. As our focus here was not on exploring whether there are specific school level (contextual) effects, but rather on the association of participation in organised 'out of hour' activities (whether that was organised by the school or a community group) on child physical activity levels, the key issue with clustering (non-independence) within schools is that conventional analyses would underestimate the standard error, meaning that 95% CIs would be narrower and p-values smaller than the correct values taking account of clustering in schools. Thus, we dealt with this issue by computing robust standard errors which take account of school level clustering and ensure that our CIs and p-values reflect the school level clustering. We have now made this clearer in the revised manuscript – please see the revised text on lines 194-197.

2. In the discussion, the authors should compare their results with those of previous studies to a greater extent. For instance, there are many previous studies (even systematic reviews) that have looked at the types of activities considered in the manuscript, such as outdoor play and after-school programs.

Response: Thank you for the helpful feedback. We have now added a paragraph which relates the findings from the current study to the previous work. This is shown on lines 310-316 and a number of references have been added to support this text.

Minor comments:

1. Was the reliability and validity of the questions used to examine the type of activities assessed in this study or in a previous study? If yes, this should be mentioned in the methods. If not, then it should be mentioned as a limitation of the study.

Response: This was a new scale that was created for this project but the reliability and validity has now been formally tested. We have now added text to recognise this limitation on lines 346-347.

2. Please justify the use of 500 minutes (8 hours and 20 minutes) as a threshold for inclusion of accelerometer data. I am asking this because it is well known that accelerometry data reduction methods vary substantially across studies which makes it difficult to compare the findings of different studies (see Cain et al. J Phys Act Health 2013;10(3):437-450 for detailed discussion of this issue).

Response: Thank you for raising this issue. We have now added the text below to lines 158 to 166 to

address this concern.

“We recognise that there is considerable variation in the number of minutes of accelerometer data that are required to be considered representative of a valid day.[26] These have ranged from 360 minutes per day which has been used for 6 to 8 year old children,[27] to 800 minutes which has been used for older children.[28 29] Within the field there is no consensus on the minimum number of minutes per day that are needed for a day to be considered valid. We, therefore, adopted a 500 minute per day threshold to ensure that our data are comparable to the methods employed by the International Children’s Accelerometer Database,[6] which has pooled data from over 27,000 children across 20 large global cohorts.”

3. The authors correctly acknowledge that they were not able to disentangle whether play in the home or garden occurred inside or outside due to the wording of the question. Another issue is that this question (and the one about play outside near the home) appeared to consider only play with friends and/or family. While there is evidence that friends/family can help encourage children to be active, it would be worthwhile to mention that the questions used do not allow the researchers to examine the contribution of play without friends and or family. Playing outside alone would likely involve more physical activity than playing inside alone.

Response: We thank the reviewer for raising this issue. We have now amended the text so that it reads as: “Equally, as the question focussed on play with friends or family we do not have any information about individual play, and we were unable to disentangle these inter-related issues.”. Please see lines 350-352.

Reviewer 2:

This is a strong study with a large sample size with four reported physical activity settings and use of accelerometers to objectively measure PA and ST. All my concerns were directly addressed in the manuscript. It is an exceptional manuscript and ready for publication.

Response: Thank you for the very supportive comment – no change has been made.